# The Critical Role of Sirt1 in Subarachnoid Hemorrhages: Mechanism and Therapeutic Considerations

**DOI:** 10.3390/brainsci13040674

**Published:** 2023-04-18

**Authors:** Zhonghua Zhang, Cong Liu, Xiaoming Zhou, Xin Zhang

**Affiliations:** 1Department of Neurosurgery, Jinling Hospital, Jinling School of Clinical Medicine, Nanjing Medical University, Nanjing 210029, China; 2Department of Ophthalmology, Jinling Hospital, Jinling School of Clinical Medicine, Nanjing Medical University, Nanjing 210029, China

**Keywords:** Sirt1, subarachnoid hemorrhage, histone acetylation, apoptosis, inflammation, oxidative stress

## Abstract

The subarachnoid hemorrhage (SAH) is an important cause of death and long-term disability worldwide. As a nicotinamide adenine dinucleotide-dependent deacetylase, silent information regulator 1 (Sirt1) is a multipotent molecule involved in many pathophysiological processes. A growing number of studies have demonstrated that Sirt1 activation may exert positive effects on SAHs by regulating inflammation, oxidative stress, apoptosis, autophagy, and ferroptosis. Thus, Sirt1 agonists may serve as potential therapeutic drugs for SAHs. In this review, we summarized the current state of our knowledge on the relationship between Sirt1 and SAHs and provided an updated overview of the downstream molecules of Sirt1 in SAHs.

## 1. Introduction

Subarachnoid hemorrhages (SAHs) are characterized by acute bleeding that rapidly accumulates in the subarachnoid space and is associated with high levels of long-term morbidity and mortality [1]. Despite significant progress in its clinical treatment, as of yet, the 30-day mortality of SAHs is 35% [2], and 40% of SAH survivors remain dependent due to physical disability, behavioral, and cognitive disturbances, which leads to high societal healthcare costs and economic impact. Different kinds of cells are involved in the pathophysiologic process of SAHs and the mechanisms following a SAH are complex and multifactorial.

Sirtuins, a family of class III histone deacetylases, have been widely investigated for their neuroprotective role, with silent information regulator 1 (Sirt1) being the most extensively studied and well-characterized member of the family. Sirt1 is present in both the nucleus and cytoplasm in several cells and can shuttle between the nucleus and cytoplasm under certain conditions [3]. As an NAD+-dependent deacetylase, Sirt1 can regulate the deacetylation of target proteins that include histones (H1K26, H3K9, H3K14, H3K18, H3K56, H4K6, H4K12, and H4K16) [4,5,6,7] or non-histone proteins {p53 [8], nuclear factor kappa B (NF-κB), Ku70 [9], forkhead box transcription factor 1/3/4 (FOXO1/3/4) [10], hypoxia-inducible factor-1a/2α (HIF-1α/2α) [11,12], proliferator-activated receptor γ coactivator1α (PGC-1α) [13], and the signal transducer and activator of transcription 3 (STAT3) [14]}. Notably, the deacetylase activity of Sirt1 is involved in multiple pathophysiological processes such as energy metabolism [15], cell differentiation [16], inflammatory response [17], oxidative stress [18], autophagy [19], and apoptosis [20]. Sirt1 also plays a confirmed role in some tumor cells, such as ovarian, thyroid, pancreatic cancers [21], and breast cancer [22]; meanwhile, Sirt1 can serve as useful biomarkers for predicting WEE1 inhibitor sensitivity or resistance [23]. It was reported that Sirt1 is highly expressed in vital brain regions that control fatal functions of cognition, memory, and motor coordination, including the hippocampus, cerebellum, and hypothalamus [24,25]. Many studies have confirmed that Sirt1 can exert a neuroprotective effect by modulating synaptic plasticity and promoting cognitive function [26,27]. Moreover, Sirt1 has been implicated in the proliferation and differentiation of neural stem cells, which is a crucial process for the development of complex neural networks in the brain [28]. In addition, the upregulation of Sirt1 expression positively affected neurogenesis and neuritogenesis via various targets and mechanisms, which makes it a potential therapeutic target to inhibit neurodegenerative disease progression [29,30]. In this review, we summarized the latest advances in the roles of Sirt1 in SAHs, with a focus on how Sirt1 regulates inflammation, oxidative stress, apoptosis, autophagy, and ferroptosis.

### The Pathological Characteristics of SAHs 

Once an intracranial aneurysm (IA) ruptures, blood entering the subarachnoid space causes the physical compression of local structures through a sharp increase in intracranial pressure (ICP), altering cerebral perfusion pressure and cerebral blood flow [31]. Notably, the resulting global cerebral ischemia and blood products will trigger many pathophysiological cascades. The outcomes of SAHs are mainly classified into early brain injury (EBI) and delayed cerebral ischemia (DCI) (Figure 1).

EBI, which occurs in the first 72 h after the bleeding, is commonly proven to be a major factor leading to adverse outcomes, and is considered a key denominator in relation to DCI and long-lasting chronic sequelae after a SAH [32]. The pathophysiologic mechanisms resulting in EBI are multifactorial and complicated. ICP increases suddenly after bleeding, followed by a decrease in cerebral perfusion pressure and cerebral blood flow, which could cause transient or persistent ischemia in some cases. Subsequently, the impairment of neurons, astrocytes, microglia, vascular smooth muscle cells (VSMCs), and endothelial cells resulted in blood–brain barrier (BBB) breakdown [33], leading to brain homeostasis disruption. Brain edema, a major cause of neurological deterioration and the death of EBI following a SAH, is mainly divided into vasogenic and cytotoxic types [34,35]. Existing evidence indicates that there is a bidirectional relationship between cytotoxic edema and vasogenic edema after a SAH. On the one hand, cytotoxic edema, characterized by cell swelling caused by a loss of energy for “pumps”, such as the Na^+^ K^+^-ATPase, the Ca2^+^-ATPase, and the uncoupling expression of Aquaporin-4 and K^+^, results in significant BBB alterations, which ends up with an abnormal accumulation of fluid in brain interstitium and aggravates the development of vasogenic edema [36]. On the other hand, vasogenic edema is associated with the extravasation of plasma proteins and the degradation of tight junction proteins, which can cause cytotoxic edema [37]. Neuroinflammation in the brain parenchyma after bleeding is a characteristic response to brain damage and likely another key component of EBI. The release of different inflammatory cytokines, among which interleukin 6 (IL-6) and matrix metalloproteinase 9 (MMP-9) play essential roles, leads to the self-reinforcement of the immune system, acting to further the brain damage [38,39].

DCI usually occurs between 3 and 14 days following aneurysmal rupture. It complicates the clinical course in approximately 20–40% of SAH patients and substantially worsens outcomes [40]. Accumulating evidence has revealed that the pathophysiology of DCI is complex and multifactorial [41]. The main mechanisms of DCI after a SAH include large vessel vasospasm, microthrombosis, microcirculatory dysfunction, cortical spreading depolarizations (SDs), impaired cerebral autoregulation, and neuroinflammation. Notably, new investigations have shown the interactions and interdependencies among pathological factors [42].

Among the complex mechanisms yielding poor outcomes following a SAH, cerebral vasospasm remains one of the main determinants of DCI, although the magnitude of this contribution is debated [43]. Cerebral vasospasm is defined as the “arterial narrowing of large cerebral vessels observed on a radiological test such as CT angiography, MRA, or digital subtraction angiography”, typically observed during the 3–14 days after a SAH [44]. The main driver of cerebral vasospasm is the release of haemoglobin from the subarachnoid clot. The multiple underlying mechanisms are poorly understood, including oxygen-based free radical reactions, the decreased production of NO, and the modification of K^+^ and Ca^2+^ channels [45]. A causal relationship between cerebral vasospasm and DCI has been questioned. On the one hand, mounting evidence has indicated a stronger correlation between cerebral vasospasm and DCI after a SAH. The abnormal narrowing or constriction of cerebral arteries was caused by blood in the subarachnoid space several days following a SAH, which could be found radiographically in 70% of SAH patients. On the other hand, DCI following a SAH was also found to occur in territories without angiographic vasospasm [46]. Furthermore, the prevention or treatment of angiographic cerebral vasospasm does not automatically lead to improved functional outcomes, as was observed in other studies [47]. Further exploration is necessary to broaden our knowledge of the role that cerebral vasospasm plays in the pathogenesis of a SAH and to facilitate the development of novel therapeutic strategies for SAHs.

## 2. The Therapeutic Role of Sirt1 in SAHs 

### 2.1. Sirt1 Regulates Inflammatory Responses in SAHs 

With accumulating evidence from both clinical research and experimental models, it is widely accepted that inflammation plays a crucial role in the numerous pathophysiological processes following the development of post-SAH complications [48]. It has been confirmed that there is a significant correlation between severe cortical inflammation and EBI after a SAH [49]. Previous studies have also indicated that inflammatory responses in the pathological course after a SAH was associated with cascade pathways, leading to the subsequent induction of proinflammatory cytokines and chemokines. The suppression of neuroinflammation in the brain by anti-inflammatory therapy prevents EBI after a SAH [50].

Microglia, the innate immune components of the central nervous system, are a crucial mediator of neuroinflammation in SAHs [51]. Several animal studies have found that microglial activation plays an important role in the development of disease complications and recovery in EBI following a SAH [52]. Targeting microglial polarization may be a potential strategy to limit neuroinflammation in early SAH management [53,54]. Similarly, according to recent single-cell analysis studies, potential interventions could target microglia-mediated neuroinflammation in SAHs [55,56].

Nucleotide-binding and oligomerization domain-like receptor pyrin domain-containing protein 3 (NLRP3), a 118 kDa cytosolic pattern recognition receptor protein, is the most extensively studied inflammasome to date due to its ability to regulate the innate immune system and inflammatory signaling [57]. Previous studies have supported the involvement of NLRP3 in the progression of a SAH; however, the mechanism remains unclear. Significant NLRP3 expression is observed in monocytes from SAH patients compared to healthy individuals. Similarly, when erythrocyte lysate stimulates monocytes, it activates NLRP3 inflammasome, which leads to the release of proinflammatory cytokines in in vitro models [58]. Moreover, the blockade or inhibition of NLPR3 inflammasome activation could attenuate early brain injury and delay cerebral vasospasm after a SAH [59,60], e.g., AMPK/Sirt1 interaction with NLRP3 inflammasome activation in an experimental mouse model of multiple sclerosis [61]. Interestingly, the depletion of ERα and Sirt1 is related to the activation of the NLRP3, facilitating the rupture of intracranial aneurysm in an aneurysm rat model under estrogen-deficient conditions [62]. Consistent with previous research, Xia et al. observed that the Sirt1 agonist could ameliorate EBI after a SAH by shifting the microglial from proinflammatory M1 to anti-inflammatory M2 state via the modulation of ROS-mediated NLRP3 inflammasome signaling [63]. Meanwhile, Dioscin, a potent activator of Sirt1, could effectively inhibit NLRP3 inflammasome activation and exert long-lasting neuroprotective effects after a SAH [64]. Previous research suggested that Sirt1 may inhibit NLRP3 and reduce neuroinflammation in SAH animal models, but the relationship between Sirt1 and NLRP3 needs to be investigated further in various aspects [65].

There is a large evidence base showing that the NLRP3-canonical pathway is produced when damage-associated molecular patterns (DAMPs) activate the NF-κB pathway via the toll-like receptors (TLRs). DAMPs, an innate immune response to molecules released from the extracellular and intracellular space of damaged tissue or dead cells, are essential in response to inflammation progression [66]. Notably, the high-mobility group protein B1 (HMGB1), a typical DAMP, plays an integral role in the regulation of neuroinflammation following a SAH [67,68]. After a SAH, HMGB1, released from necrotic neurons, could induce cytokine release and leukocyte recruitment and then trigger the inflammatory response after interaction with TLR4 and the receptor for advanced glycation end products (RAGEs) [69]. Experimental animal studies have revealed that the inhibition of HMGB1 could mitigate an inflammatory response and ameliorate EBI after a SAH [70,71,72]. Moreover, clinical research suggests that the serum HMGB1 of SAH patients may serve as an independent biomarker predictive of DCI [73]. Another study has found that HMGB1 levels in cerebrospinal fluid are correlated with treatment outcomes in patients with acute hydrocephalus following a SAH [74]. HMGB1 release is closely associated with its acetylation level when undergoing acetylation modification, HMGB1 shuttle from the nucleus to the cytoplasm, and promoted downstream inflammatory signal transduction [75]. Meanwhile, Sirt1 can directly interact with and deacetylate HMGB1 and eventually inhibit HMGB1 activity in inflammation regulation [76]. Therefore, the regulation of the Sirt1-HMGB1 axis has been proposed as an aspect of the anti-inflammatory role to alleviate EBI after a SAH. For example, oleanolic acid exerts neuroprotective effects in SAH rats via Sirt1-mediated HMGB1 deacetylation [77].

The inhibition of NF-κB activity is another effective method associated with the Sirt1-dependent inhibition of neuroinflammation. There is a relationship between extracellular HMGB1 and toll-like receptor 4 (TLR4) via the medullary differentiation factor 88 (Myd88)-dependent pathway, aggravating NF-kB-dependent proinflammatory mediator secretion in an in vivo rat SAH model. The activation of Sirt1 decreased the nucleocytoplasmic translocation of HMGB1 and inhibited the TLR4/NF-кB pathway, thereby ameliorating the subsequent induction of proinflammatory cytokines and secondary brain injury after a SAH [78]. Intriguingly, some researchers have indicated the suppression of SAH-triggered cerebral inflammation alleviation neurological dysfunction via the Sirt1/NF-κB pathway [79,80]. In another recent study, Berberine, a potential Sirt1 activator, ameliorated the state of neuroinflammation and subsequently improved the neurological behavior in mice through the Sirt1-modulated downregulation of HMGB1/NF-κB activity. EX527, a Sirt1 inhibitor, reversed the anti-inflammatory and neuroprotective effects [81]. Considering the previous findings, it can be speculated that Sirt1 acts as a potent target for inducing anti-inflammatory and neuroprotective effects (Table 1).

### 2.2. Sirt1 Regulates Mitochondrial Function and Oxidative Stress in SAHs

Mounting evidence has shown that oxidative stress plays an essential role in the EBI and consequently in the DCI of SAHs. Mitochondria, i.e., double-membrane organelles, are the main source of energy production via biological oxidation in eukaryotic cells. It is well established that SAH-induced mitochondrial dysfunction, which can lead to the overproduction of reactive oxygen species (ROS), is closely associated with oxidative stress in EBI and DCI after a SAH [82]. Previous studies have reported that a higher mitochondrial membrane potential in the cerebrospinal fluid (CSF) of SAH patients is closely related to good outcomes [83]. Another study has also discovered extracellular mitochondrial dysfunction in the CSF of patients with DCI after a SAH [84].

Mitochondrial biogenesis, which can maintain mitochondrial homeostasis during the mitochondrial life cycle, is considered an essential mechanism of the reversion of mitochondrial dysfunction and the reduction in ROS production. An experimental mice study has revealed that promoting mitochondrial biogenesis contributes greatly to neuroprotection after a SAH [85]. The peroxisome proliferator-activated receptor gamma coactivator 1-alpha (PGC-1α) pathway plays an essential role in a regulatory network that governs the transcriptional control of mitochondrial biogenesis via targeting multiple transcription factors, including nuclear respiratory factor 1/2 (NRF-1/2) and estrogen-related receptor α (ERRα) [86,87]. Furthermore, it is noticeable that the NAD+/NADH ratio plays a pivotal role in the regulation of mitochondrial biogenesis. In response to NAD+, Sirt1 deacetylates multiple lysine residues in PGC-1α, thus leading to its activation [88]. Resveratrol, an activator of Sirt1, reduced mitochondrial dysfunction-induced oxidative stress following a SAH by improving mitochondrial biogenesis via the PGC-1α pathway [89]. Another study has also found that regulating the AMPK/Sirt1/PGC-1α signaling pathway could alleviate intracellular oxidative stress and participate in neuroprotection, thereby ameliorating EBI after a SAH [90].

Forkhead box protein O1 (FoxO1), the most widely studied subtype of the FoxO family, is an important regulator of oxidative defense, such as catalase (CAT) and superoxide dismutase (SOD) expression, thereby maintaning biological redox homeostasis. Accumulating experimental evidence has found that FoxO1 acetylation increased significantly in the brain after SAH insults. Interestingly, acetylation can regulate the transcriptional activity of FoxO1. A recent study found that the deacetylation of FoxO1 plays a neuroprotective role against oxidative stress after cerebral ischemia-reperfusion [91]. Sirt1, an upstream signaling mediator of FoxO1, could deacetylate FoxO1 to alter FoxO1 DNA binding affinity and promoter binding specificity, which further induces cellular resistance against oxidative damage and participates in neuroprotection after a SAH [92,93,94].

Considerable evidence has shown that the pathway involving nuclear factor erythroid 2-related factor 2 (Nrf2), a vital endogenous antioxidant system, may be an effective protective mechanism for EBI following a SAH [95,96]. The beneficial role of Nrf2 in maintaining redox homeostasis has been proven to be associated in vitro with Sirt1. On the one hand, Sirt1 could increase Nrf2 transcription, thus leading to increases in its protein levels; on the other hand, Sirt1 could decrease the ghd polyubiquitination of Nrf2 by decreasing the expression of Keap1/Cul3 and increasing Nrf2’s ability to bind to antioxidant-responsive elements (AREs) [97]. As an antioxidant therapy, a powerful Sirt1 activator, isoliquiritigenin, could increase the activation of Nrf2 and alleviate SAH-induced oxidative damage, and has a prominent neuroprotective effect [98]. Similarly, our team showed that salvianolic acid B, a potent activator of Sirt1, dramatically reduced neurologic impairment and brain edema by upregulating the Nrf2 signaling pathway after a SAH. Notably, the Sirt1-specific inhibitor sirtinol reversed the therapeutic effects of salvianolic acid B against a SAH by inhibiting Sirt1 as well as Nrf2 activation [99]. Considering the findings above, Sirt1 could act as a potent target for antioxidant therapy (Table 2).

### 2.3. Sirt1 Regulates Apoptosis in the Pathophysiology of SAHs

Multiple types of programmed cell death (PCD), including apoptosis, pyroptosis, autophagy, necrosis, and ferroptosis, have been determined to be involved in EBI. Apoptosis, the first and best characterized type of PCD, undergoes a variety of signal transductions and maintains the homeostatic balance between cell survival and cell death under normal conditions. In the pathogenesis of SAHs, it has been found that, in addition to necrosis, cell apoptosis plays a pivotal role in SAH neuronal loss. An increasing number of experimental studies have supported the view that apoptosis can occur in cortical, subcortical, and hippocampal neurons and the endothelium after a SAH [100]. Apoptosis can be partitioned into the following pathways: the death-receptor-dependent pathway (the extrinsic pathway), the mitochondria-dependent pathway (the intrinsic pathway), and the endoplasmic reticulum (ER)-stress-induced pathway. Several apoptotic pathways are activated after a SAH, and targeting the upstream and downstream molecules of apoptosis-related pathways are effective therapeutic strategies against EBI [101].

Sirt1 functions in SAH-induced neuronal apoptosis and neuroprotective effects have been discovered [102]. The results obtained by Qian et al. demonstrated that, in a rat SAH model, Sirt1 overexpression upregulated the expressions of B-cell lymphoma-2 (BCL-2), tight junction proteins ZO-1, occludin, and Claudin5, and downregulated the expression of p53, Bax, and cleaved caspase-3, suggesting that Sirt1 could ease brain edema and decrease apoptosis which is closely related with death rate and neurological function after a SAH [103]. Similarly, another study also confirmed that Sirt1-mediated p66shc suppression inhibited apoptosis, as measured by immunofluorescence, Fluoro-Jade C (FJC), and terminal deoxynucleotidyl transferase dUTP nick-end labeling (TUNEL) staining [104]. In addition, Cheng et al. showed that Sirt1 activation and p53 downregulation significantly reduced SAH-mediated apoptosis [105].

Akt (also known as protein kinase B) is a key antiapoptotic signaling downstream of phosphoinositide 3-kinase (PI3K) that participates in a broad range of cellular processes. Mounting evidence has indicated that the activation of the Akt signaling pathway significantly improved neurological and cognitive functional outcomes [106] and contributed to the reduction in EBI and neuronal apoptosis after a SAH [107]. It is well established that Sirt1 interacts with Akt [108]. Previous studies have shown that there was an interaction between Sirt1 and the upstream phosphatase and tensin homolog (PTEN) of Akt [109]. Meanwhile, Sirt1 can increase the membrane localization and activation of Akt via the deacetylation of Akt [110]. Existing evidence indicates that EBI following a SAH could be attenuated by suppressing neuronal apoptosis through the Sirt1/Akt pathway [111]. Even so, the way in which Sirt1 regulates the Akt signaling pathway has not been fully elucidated, and further research is urgently needed to decipher the underlying patterns.

Existing evidence has indicated that NF-κB regulates the activity of apoptotic mediators in SAH-induced EBI, and previous studies have suggested that NF-κB could be a potential therapeutic target in EBI prevention [112]. An experimental study of adult mice has demonstrated that inhibiting the NF-κB signaling pathway by inhibiting the phosphorylation of transforming growth factor-β-activated kinase 1 (TAK1) effectively reduced neuronal apoptosis, thereby ameliorating neurological deficits and improving motor coordination in SAHs [113]. Additionally, the results obtained by Zhao et al. demonstrated that Sirt1/NF-κB signaling pathway mediates the protective effect of Melatonin against EBI following a SAH in mice [114]. These results manifestly demonstrate that the regulation of Sirt1 could reduce neuronal apoptosis in SAH models (Table 3). However, because the apoptotic pathway that is associated with the effects of Sirt1 on apoptosis in SAHs is not completely understood, further research is urgently needed to decipher the underlying patterns.

### 2.4. Sirt1 Regulates Autophagy in the Pathophysiology of SAHs

Autophagy, a beneficial catabolic process maintaining the cellular and tissue homeostasis under various forms of stress, has been proven to be neuroprotective [115]. After an experimental SAH, autophagy was activated during the early stages, peaked at 24 h, and lasted up to 48 h in the rat brain [116]. Emerging evidence has revealed that modulating autophagy protected against SAH-induced brain injury [117]. Our recent study found that Acetyl CoA synthase 2 improved neurological deficits, brain edema, and neuronal death, which are shown to be responsible for poor outcomes after a SAH by enhancing autophagy [63], consistent with the results of Hao et al. [118]. The activation of autophagy may be a promising strategy in the treatment of SAHs.

Acetylation has been previously demonstrated as an essential regulatory mechanism for autophagy, regulating autophagy initiation and autophagosome formation [119]. As the major deacetylase functioning in autophagy, Sirt1 has been shown to interact with essential components of the autophagy machinery [120]. Specifically, Sirt1 acts as a critical energy sensor and promotes autophagic responses by the deacetylation of several autophagy-related (ATG) proteins, such as ATG5, ATG6, ATG7, and ATG8/light-chain microtubule-associated protein (LC3) under nutrient deprivation conditions [121]. Sirt1 can deacetylate nuclear LC3, a key marker of autophagosomal membranes, to facilitate autophagy initiation [122]. Sirt1 exerts neuroprotective effects via the regulation of autophagy in various neurological disorders. The previous study indicated that Sirt1 could enhance macroautophagy in the astrocytes by upregulating LC3 expression to improve functional recovery after brain injury [123]. Similarly, another study demonstrates that queen bee acid can exert a neuroprotective effect in PD models through the deacetylation of LC3 and BECN1 proteins mediated by Sirt1 activation. [124] More recently, an experimental study reported that Sirt1 potently promoted chaperone-mediated autophagy (CMA) activity, the third type of autophagy, which has been identified as an important process for nerve injury and contributes to neuroprotection after closed head injury by modulating the deacetylation and ubiquitination of molecular chaperone DnaJ heat shock protein family member B1(Dnajb1) [125].

AMPK, an evolutionarily conserved serine–threonine protein kinase, plays a major role in the control of energy metabolism. AMPK activation, which is precisely regulated by the ratio of AMP/ATP and ADP/ATP in the cells, is closely associated with the activity of Sirt1 by increasing cellular NAD + levels. There are several lines of evidence supporting the fact that the activation of autophagy could be regulated by the AMPK/Sirt1 pathway [126,127]. Li et al. observed that resveratrol attenuated the release of pro-inflammatory cytokines and neural apoptosis after a SAH, and that the protective effects of resveratrol on EBI following a SAH are mediated via the activation of the AMPK/Sirt1/autophagy signaling pathway [128] (Table 3). However, the detailed mechanism by which AMPK/Sirt1 regulates autophagy activation in SAHs remains unclear.

### 2.5. Sirt1 Regulates Ferroptosis in the Pathophysiology of SAHs

Ferroptosis is a newly identified oxidative form of programmed cell death caused by lipid peroxidation accumulation and membrane damage in eukaryotic organisms, accompanied by iron overload [129]. The morphological features of ferroptosis mainly include outer membrane rupture and blistering, mitochondrial membrane density compression, and a reduction in mitochondria crista which is a unique sign that distinguishes it from other forms of death [130].

The functions of ferroptosis in SAHs have been studied. The results obtained by Li et al. demonstrated that the ferroptosis inhibitor ferrostatin-1 (Fer-1) significantly ameliorated neuronal death in the oxyhemoglobin-induced neuron injury model, as demonstrated by the MTT (3-(4,5-dimethylthiazol-2-yl)-2,5-diphenyltetrazolium bromide) assay, the lactate dehydrogenase (LDH) release assay, and FJC staining [99]. In addition, Kuang et al. showed that in a rat SAH model, the inhibition of ferroptosis by suppressing lipid peroxidation has been recently reported to protectively reduce blood–brain barrier impairment, brain edema, behavioral deficits, and neuronal damage after a SAH [131]. In another study conducted by Cao et al., they found that the selective ferroptosis inhibitor liproxstatin-1 provided neuroprotection against EBI after a SAH, reducing the activation of microglia and alleviating neuroinflammation, as shown by the decreased expression of IL-6, IL-1β, and tumor necrosis factor α (TNF-α) [132]. In conclusion, these data indicate that ferroptosis plays a detrimental role in the EBI of SAHs in vitro and in vivo [132].

Numerous studies have proposed the functions of Sirt1 in ferroptosis. Liu et al. found that pumilio 2 (PUM2) aggravated the neuroinflammation and brain damage induced by ischemia–reperfusion through ferroptosis by inhibiting the Sirt1 [133]. Moreover, Li et al. [134] proposed that Sirt1 upregulation attenuated ferroptosis levels in ischemic ischemic hypoxic–ischemic brain injury and improved learning and memory via the Nrf2/GPx4 signaling pathway. Our recent study found that Sirt1 is hardly expressed and that a higher expression of Sirt1 protects brain tissues after SAHs by alleviating ferroptosis, which provides evidence for its potential use as a therapeutic target [135] (Table 3).

### 2.6. Sirt1 Regulates the Neuroprotection of Hypoxic Postconditioning in SAHs

It is well established that initial subarachnoid bleeding is a major contributor of early brain injury. It is also of note that subsequent neurological deterioration from DCI remains a pivotal part of preventable morbidity and mortality. Hypoxic postconditioning, an endogenous cellular protection mechanism, refers that sublethal hypoxic exposure can increase the resistance of cells, tissues, or organs to subsequent lethal stimulation [136]. The molecular signaling cascades of endogenous brain protection are being identified [137].

Experimental data have shown that hypoxic postconditioning initiated at clinically relevant time points after a SAH provides strong protection against cerebral vasospasm, microvessel thrombi, and neurological deficits; this protection is multifaceted, acting at both macro- and microvascular levels. A recent study has highlighted that Sirt1 is a mediator of the strong endothelial nitric oxide synthase-mediated neurovascular protection against a SAH afforded by hypoxic preconditioning. Hypoxic preconditioning leads to a rapid and sustained increase in cerebral Sirt1 mRNA and protein expression. Additionally, the Sirt1-activating polyphenol resveratrol could mimic the DCI protection afforded by hypoxic preconditioning [138]. More recently, Diwan and colleagues using Sirt1 knockout and Sirt1-overexpressing mice provided cross-validating genetic and pharmacologic evidence that Sirt1 mediates protection against DCI in SAHs in response to hypoxic postconditioning [139]. Emerging animal studies have demonstrated that postconditioning with Sirt1-promoting therapeutics represents a clinically viable treatment strategy for reducing secondary brain injury and improving overall outcomes in patients presenting with SAHs and further research is required to confirm the validity and reliability of these treatments.

## 3. Conclusions

Up to now, the therapeutic options available for SAHs are limited; thereby, the identification of new therapeutic strategies is urgently needed. Meanwhile, the neuroprotective roles of Sirt1, an NAD+-dependent protein deacetylase that has been traditionally linked with calorie restriction and aging, in the context of cerebral ischemia and neurodegenerative disorders, have attracted more and more attention. As a new direction for the therapeutic approach for SAHs, Sirt1 could play important roles im various ways (Figure 2), e.g., by suppressing the inflammatory response, mitigating oxidative damage, alleviating apoptosis, accumulating the autophagy, and inhibiting ferroptosis. In this review, we summarize the functions as well as some downstream moleculars of Sirt1 in SAHs. Both in vitro and in vivo studies showed that it may exert a protective role after a SAH via multiple mechanisms, although the case of whether the known or currently unidentified pathways are most important remains unclear. Moreover, the value of its clinical application demands deeper exploration. Meanwhile, in current laboratory testing, the use of Sirt1 activators could still be an innovative new therapy for patients with SAHs.

## Figures and Tables

**Figure 1 brainsci-13-00674-f001:**
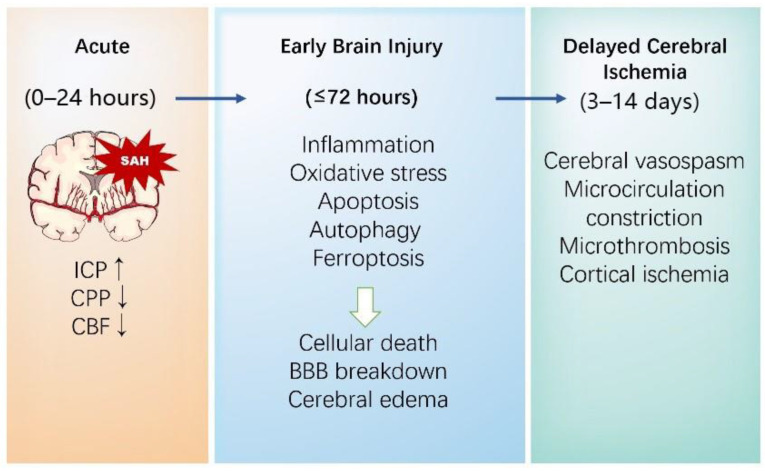
The pathophysiological mechanisms of brain injury following a SAH.

**Figure 2 brainsci-13-00674-f002:**
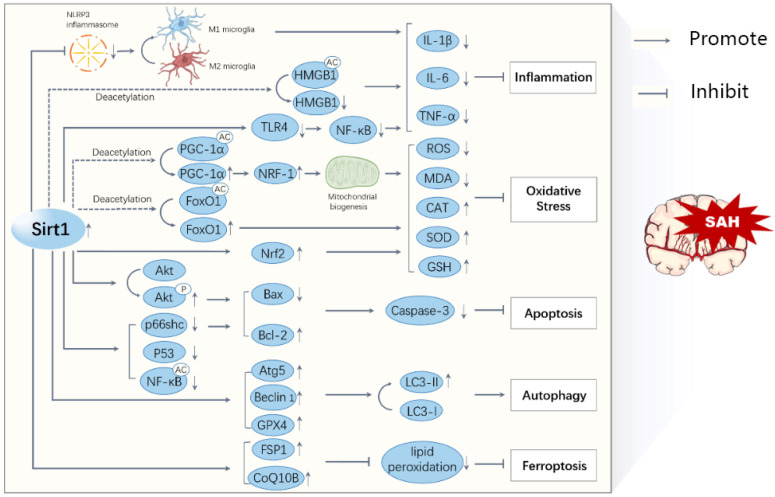
A schematic diagram illustrating the major mechanisms of Sirt1 in SAHs. The activation of Sirt1 reduces the acetylation and activation of transcription factors, such as HMGB1, PGC-1α, FoxO1, and NF-kB, leading to decreased inflammation, oxidative stress, apoptosis, ferroptosis, and increased autophagy. All these processes interact with each other and contribute to the progression of a SAH.

**Table 1 brainsci-13-00674-t001:** Related studies exploring Sirt1-targeted anti-inflammation therapy for SAHs.

Studied Drugs	Mechanisms/Beneficial Functions of the Regulation of Sirt1	Related Study Results	Animals and/or Cells	References
SRT1720	Suppress the inflammatory response, oxidative damage, and neuronal degeneration	SRT1720 decreased IL-1b, IL-6, TNF-a, IL-18, ICAM-1, CCL-2, NLRP3, and ASC; cleaved caspase1 protein levels	Rats, rat cortical neurons	[63]
Dioscin	Inhibit the inflammatory response, oxidative damage, neurological impairment, and neural cell degeneration	Dioscin decreased caspase-3, Bax, and P53, and increased protein Bcl2	Mice, mice cortical neurons	[64]
Oleanolic acid	Suppress inflammation; improve the grading score, the neurological score, brain edema, and permeability of the brain–blood barrier	Oleanolic acid decreased the acetylation level of HMGB1; inhibited the expression of TLR4, the degradation of IκBα, NF-κB p65 nuclear translocation, and IL-1β and TNF-α	rats	[77]
Magnesium Lithospermate B	Attenuate brain edema and neurological deficits, inhibit the activation of microglia, and reduce neuronal apoptosis	Magnesium; ithospermate B increased the expression of SIRT1, inhibited the acetylation of NF-κB, decreased the expression of TNF-α, and cleaved caspase-3	rats	[79]
Astaxanthin	Ameliorate cerebral inflammation, brain edema, and neuronal death; improve neurologic function	Astaxanthin inhibited HMGB1, TLR4, MyD88, NF-кB p65, IL-1b, TNF-a, and ICAM-1; cleaved caspase-3 and Bax expression; and enhanced level of Bcl2.	Rats, mice, TLR4 gene KO mice, mice cortical neurons, and microglia	[78]
Rolipram	Ameliorate brain edema and alleviate neurological dysfunction	Rolipram promoted the expression of Sirt1; inhibited NF-κB activation; inhibited the activation of microglia; down-regulated the expression of TNF-α, IL-1ß, and IL-6; and increased the expression of IL-10	rats	[80]
Berberine	Improve neurological behavior, reduce brain edema, attenuate inflammation, and decrease neural apoptosis	Berberine decreased Sirt1 expression; increased protein levels of HMGB1, TLR4, Myd88, and Nf-kB p65; inhibited microglia activation; decreased IL-1β, IL-6, TNF-a, ICAM-1, and caspase-3 levels	rats	[81]

**Table 2 brainsci-13-00674-t002:** Related studies exploring Sirt1-targeted antioxidant therapy for SAHs.

Studied Drugs	Mechanisms/Beneficial Functions of the Regulation of Sirt1	Related Study Results	Animals and/or Cells	References
BMS-470539	Attenuate neurological deficits; reduce long-term spatial learning and memory impairments; suppress oxidative stress, apoptosis, and mitochondrial fission	BMS-470539 increased Sirt1, PGC-1α, UCP2, SOD, GPx, Bcl-2, cyto-Drp1, and ATP levels; decreased cleaved caspase-3, Bax, mito-Drp1, ROS, and GSH/GSSG levels, as well as NADPH/NADP+ ratios	Rats	[90]
Fucoxanthin	Improve neurological function, reduce brain edema, ameliorate neurodegeneration, and mitigate oxidative damage	Fucoxanthin decreased lipid peroxidation, nitrotyrosine, and 8-OHdG production and increased SOD, GSH, GSH-Px, and CAT	Rats, rat cortical neurons	[92]
Isoliquiritigenin	Reduce brain edema, improve behavioral function, ameliorate neuronal degeneration, and suppress suppresssuppresses oxidative damage after a SAH	Isoliquiritigenin induced Sirt1; Nrf2 activation decreased MDA levels and ROS contents, and increased SOD, GSH, and GSH-px levels and CAT activities	Rats, mice cortical neurons	[98]
Salvianolic acid B	Suppress oxidative stress; reduce neurologic impairment, brain edema, and neural cell apoptosis	Salvianolic acid B suppressed reactive oxygen species generation; decreased lipid peroxidation; and increased glutathione peroxidase, glutathione, and superoxide dismutase activities	Rats, Nrf2 KO mice, mice cortical neurons	[99]

**Table 3 brainsci-13-00674-t003:** Related studies exploring Sirt1-targeted PCD therapy for SAHs.

Studied Drugs	Mechanisms/Beneficial Functions of the Regulation of Sirt1	Related Study Results	Animals and/or Cells	References
resveratrol	Decrease apoptosis, reduce the brain water content, and improve dysfunctional BBB permeability	Resveratrol increased ZO-1 and occludin expression, and Claudin5 decreased p53 and AC-p53 expression, and lowered Bax, Puma, Noxa, and Bid mRNA expression levels	rats	[103]
Carnosic Acid	Ameliorate brain edema and BBB disruption; decrease apoptosis	Carnosic acid increased Sirt1, MnSOD, and Bcl-2 expression, and also decreased p66shc, Bax, and cleaved caspase-3 expression	Rats, PC12 cells	[104]
Wogonoside	Suppress SAH-induced edema and neuronal apoptosis	Wogonoside reduced the SAH-mediated promotion of Bax, Puma, Noxa, and Bid, and cleaved Caspase-3 expression.	rats	[105]
Phosphodiesterase-4	Decrease apoptosis; attenuate brain edema and neurological dysfunction	Rolipram increased the expression of Sirt1 and up-regulated the phosphorylation of Akt after a SAH	Rats	[111]
Melatonin	Improve neurological deficits; reduce the brain water content and neuronal apoptosis	Melatonin enhanced the expression of Sirt1 and Bcl-2 and decreased the expression of Ac-NF-κB and Bax	mice	[114]
Resveratrol	Affect autophagy; attenuate neural apoptosis and inflammation	Resveratrol increased the LC3-II/I ratio and phosphorylation state of AMPK and SIRT1 protein expression in brain tissues	rats, rat cortical neurons	[128]
resveratrol	Alleviate ferroptosis	Sirt1 activation could suppress SAH-induced ferroptosis by upregulating the expression of glutathione peroxidase 4 (GPX4) and ferroptosis suppressor protein 1 (FSP1)	mice, HT-22 cells	[135]

## Data Availability

Not applicable.

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
