# Peer review of "The Critical Role of Sirt1 in Subarachnoid Hemorrhages: Mechanism and Therapeutic Considerations"

_brainsci, 2023, doi:10.3390/brainsci13040674_

Round 1

Reviewer 1 Report

Comments and Suggestions for Authors

Dear Authors,

please, address the following issues.

Major issues

Figures: one figure is not enough for this type of Review article.  Please, prepare more artwork.

Line 40: more references are needed, since the sentence contains several transcription factors and mediators that have been extensively described in the literature.

Lines 40-43: cancer is a disease where SIRT1 involvement has been described. Specific references must be included. Please comment the specific references:   Jiang Y Oncogene 2023 PMID: 36690678. Zhu X Nat Chem Biol 2023 PMID: 36635566. Frazzi R Front Endocrinol (Lausanne) 2018 PMID: 30319549.

Lines 67-69: the sentence is not understandable. Add the "s" at the end of each verb (ICP raises ...- blood flow decreases ...) and rephrase the sentence. 

Line 85: the reference(s) is missing at the end of the sentence.

Lines 99-101: the sentence is not understandable.

Please, check the manuscript for English style and grammar, having a professional proofreading available.

Minor issues.

Table I: the references numbers should be formatted according to the text, following the order in which they appear.

Reviewer 2 Report

Comments and Suggestions for Authors

Review to the article “The Critical Role of Sirt1 in Subarachnoid Hemorrhage: Mechanism and Therapeutic Considerations” (Manuscript ID: brainsci-2269382)

Thank you for the possibility for reviewing the manuscript.

Zhang et al. present a review about the role of Silent information regulator 1 (Sirt1) in subarachnoid hemorrhage patients. As a nicotinamide adenine dinucleotide-dependent deacetylase, Sirt1 is a multipotent molecule involved in many pathophysiological processes. Several studies indicate that activation of Sirt1 may exert positive effects on SAH by regulating inflammation, oxidative stress, apoptosis, autophagy, and ferroptosis and therefore Sirt1 agonists could be helpful in treatment of SAH patients. The authors present a review about the relationship between Sirt1 and SAH and provided an updated overview of downstream molecules of Sirt1 in SAH. The authors identified that in vitro as well as in vivo studies showed that it may exert a protective role after SAH via multiple mechanisms, although which of the known or as yet unidentified pathways is most important remains unclear.

The review is structured and well written. There are only a few requests to the authors:

1)    The authors presents the benefits of Sirt 1, however they should also present an opportunity of its clinical implementation and application?

2)    In addition, the authors should suggest the next step of potential studies concerning a clinical implementation

Reviewer 3 Report

Comments and Suggestions for Authors

In this review article by Zhang et al. presents interesting information regarding role of Sirt1 in subarachnoid hemorrhage. There are some concerns that must be clarified further by the authors:

1) The authors neglected to consider the important role of vasospasm of cerebral arteries in subarachnoid hemorrhage. In this disease, abnormal narrowing or constriction of cerebral arteries are caused by blood in the subarachnoid space. The authors need to add a new paragraph describing the mechanisms of vasospasm of cerebral arteries.

2) The authors need to include important evidence of cerebral ischemia protection by SIRT1 based on studies in Sirt1 −/− mice (Diwan D. et al. J Am Heart Assoc. 2021 Oct 19; 10(20): e021113).  Moreover, hypoxic preconditioning (that attenuates vasospasm and neurological deficits after subarachnoid hemorrhage) leads to rapid and sustained increase in cerebral SIRT1 mRNA and protein expression in vivo (Vellimana Ak. et al. Exp Neurol. 2020 Dec;334:113484).

3) Table 1 should be split in several parts and integrated into paragraphs 2.1.-2.5. where there is a detailed description of studies present.

Round 2

Reviewer 1 Report

Comments and Suggestions for Authors

Dear Authors, please, address these issues:

Lines 37-40: different brackets should be put at the beginning and at the end of the period, since round and square brackets are already present in the middle. I suggest { } .

Line 42: "energy metabolism" is repeated twice. Also, the references cited at the end of the period (line 43) do not refer to cell differentiation, inflammatory response or oxidative stress... that are contained in the text !! The correct references should be put at the end of this phrase. A separate phrase should mention the demonstrated role of SIRT1 in cancer and all the three references suggested in the first round of revision must be inserted.

Please, check for minor spelling errors, missing spaces (line 183, before Table 1) etc.

Author Response

We sincerely thank you for your valuable suggestions to improve the quality of our manuscript. According to your nice suggestions, we have made extensive corrections to our previous draft, the detailed corrections are listed below. If there are any other modifications we could make, we would like very much to modify them and we really appreciate your help.

Point 1:

Lines 37-40: different brackets should be put at the beginning and at the end of the period, since round and square brackets are already present in the middle. I suggest { } .

Response 1: We sincerely thank the reviewer for careful reading. As suggested by the reviewer, we have corrected the “()” into “{ }”. (Lines 37-40, Page 1)

Point 2:

Line 42: "energy metabolism" is repeated twice. Also, the references cited at the end of the period (line 43) do not refer to cell differentiation, inflammatory response or oxidative stress... that are contained in the text !! The correct references should be put at the end of this phrase. A separate phrase should mention the demonstrated role of SIRT1 in cancer and all the three references suggested in the first round of revision must be inserted.

Response 2: Thanks for your kind suggestions, which is valuable for improving the manuscript.We have checked the literature carefully and added all the three references suggested in the first round of revision in the revised manuscript. (Line 43-46,Page 1)

Point 3:

Please, check for minor spelling errors, missing spaces (line 183, before Table 1) etc.

Response 3: We feel sorry for our carelessness. In our resubmitted manuscript, the missing spaces is revised. Thanks for your correction.

Reviewer 3 Report

Comments and Suggestions for Authors

No further comments.

Author Response

We feel great thanks for your professional review work on our article.